# Internationalization, cultural appreciation and institutional governmentality for quality control in transnational higher education cooperation: An empirical assessment

Jinsheng (Jason) Zhu[1,2], Shushu Wang[3]*

1 Scholar, Belt and Road International School, Guilin Tourism University, Guilin, Guangxi, China, 2 Visiting Scholar, School of Politics and Public Administration, Guangxi Normal University, Guilin, Guangxi, China, 3 Scholar, International Exchange & Cooperation Department, Guilin Tourism University, Guilin, Guangxi, China

* ceciliawang.gtu@qq.com

## Abstract

This article examines the dynamic mechanism of cultural appreciation and institutional governmentality to ensure successful quality control in a transnational higher education collaboration context. Adopting participatory action research and a case study approach, this paper investigates the quality control system in a Chinese tourism university. The present study finds that mutual cultural appreciation, responsible government guidance and institutional governmentality are essential quality control measures for transnational higher education cooperation. The quality control system is suggested to be established to enrich and improve the quality standards of joint international higher education collaboration. This study proposes to expand the international influence and recognition of China-foreign education collaboration through quality international exchange and cooperation.

## Introduction

There is no doubt that there is rising enthusiasm among higher education institutions across nations of the globe to build the transnational higher education setting to meet developing global demands. As it is observed, the internationalization of higher education is experiencing a prominent changing scenario [1–4]. Heterogeneous internationalization modes have resulted in altering workplace environments [1], various viewpoints [5], shifting rationales [6], theoretical and strategic management [7], as well as observational possibilities and challenges [8]. Historically, the concept of internationalization of higher education began with academics travelling to academic centres around the world to study wherever they could [9]. In recent decades, there has been a growing emphasis on the international components of higher education at institutional policies and international faculties in international higher education institutions [10]. Furthermore, there is an early separation between market-driven interests in the recruitment of fee-paying overseas students and a cost recovery industry for a marginal activity aimed to support globalization in general [11, 12]. A rising number of practitioners perceive internationalization efforts as a strategy of boosting personal and professional growth as a

**Data Availability Statement:** All relevant data are included in the paper.

**Funding:** The following phrases are included in the article's acknowledgment section. "In the article acknowledgement, we state the following words. "This article is part of academic achievements of first-class universities and disciplines in tourism management discipline (project) in Guangxi, China. The corresponding author has also been participating in research projects supported by Guilin Tourism University-China ASEAN Research Centre. This research project is financially supported by Guangxi Tourism Vocational Education Teaching Steering Committee - 2021 Tourism Vocational Education Research Project on Teaching Reform in Tourism Education (2021LYHZWZ001)." The "first-class universities and disciplines in tourism management discipline (project) in Guangxi, China" was a project of the university that the authors are working in. The Guilin Tourism University-China ASEAN Research Centre is one of the research centers that the authors Dr. Jinsheng (Jason) Zhu are working with. The "Guangxi Tourism Vocational Education Teaching Steering Committee - 2021 Tourism Vocational Education Research Project on Teaching Reform in Tourism Education" is one of the research projects undertaken by Dr. Jinsheng (Jason) Zhu. Here we would like to restate that both the authors are working and taking salaries in Guilin Tourism University, the institution we disclosed in the author title page. However, "The university and the funders had no role in study design, data collection and analysis, decision to publish, or preparation of the manuscript".

**Competing interests:** The authors have declared that no competing interests exist.

means of changing the system [13]. More subsequently, the word "internationalization" has developed and gained popularity as a fashionable worldwide phenomenon [14, 15] and has been linked to the United Nations' Sustainable Development Goals [16].

Transnational higher education has grown and attracted growing attention globally as globalisation has deepened in the second part of the twentieth century [17]. China-foreign cooperative education, as a manifestation of transnational higher education in China, has grown significantly in recent years and is becoming a crucial subject in higher education research [18]. King [19] investigates China's distinctive traits and particularities of internationalization challenges in international higher education, notably its fast expansion in international collaboration with African nations. From 71 in 1995 [20] to 2238 in 2019 [21], the number of China-Foreign Cooperative Education Programs between Chinese institutions and foreign universities has increased dramatically. Nevertheless, while the Chinese government supports internationalization of higher education as a means of improving domestic higher education's global rankings, the surge of China-foreign cooperation programs has been accompanied by quality issues raised by the introduction of mediocre foreign institutions driven by profits, as well as a lack of well-established legislation and regulations, inner governance systems, and quality control systems [22]. As just an outcome, in addition to the updating of legislation and regulation perspectives at the national level, initiatives undertaken by cooperation institution partners are the true engine for the actual running of cross-border education, ultimately promoting teaching quality and protecting the interests of all stakeholders and deserve more attention. Starting with a case study, this paper examines quality control in transnational cooperation from cultural and institutional governmentality perspectives, attempting to identify significant measures to ensure the long-term and potential promotion of teaching quality in transnationally cooperative education.

The instance selected for this research is an applied tourism university in a world-renowned tourist location in China's southwest (hereinafter GTUC). GTUC attempts to conduct a China-foreign education cooperation program to introduce the elite hospitality education resources of one University for hospitality education from Switzerland (hereinafter EHLS), which is one of the world's first institutions of hospitality management and ranks top throughout this field worldwide, in order to meet the increasing needs of the local market for high-end international hospitality professionals. Since 2015, an independent teaching and training facility has been built as the foundation of the Faculty of GL (An International Hospitality Management School) (hence GTUC-GL, or GL for abbreviation), which is a GTUC school specifically designed to conduct this program and to be the centre of this research. GTUC has been approved by the Ministry of Education China to enrol bachelor students in this special hospitality management since 2017, after formally entering the certified collaboration with EHLS in 2016. The current study aims to demonstrate the importance of cultural appreciation and institutional governmentality in quality control in transnational education cooperation through this case study, and it proposes to expand the international influence and recognition of China-foreign education collaboration through quality international exchange and cooperation. Following a review of previous research, the paper examines the cultural appreciation reflected in GTUC-hospitality GL's mindset, institutional governmentality, and quality control process in GL as a case study to discuss conceptual quality control in transnational higher education cooperation programs.

## Literature review

Before delving into the challenges posed by the globalization of higher education, we would want to clarify the definitions of important terminology used in this study. The international

curriculum, as defined by the OECD's Centre for Educational Research and Innovation (CERI), is "an international orientation in content, aimed at preparing students for performing (professionally/socially) in an international and multicultural context, and designed for both domestic and foreign students" (p9) [23]. In light of this, the following aspects will be investigated in this section of the literature as part of the present article: what are the theoretical foundations of internationalization and transnational higher education; what is currently known in the field; and what are the existing gaps throughout the concepts? In order to provide a more comprehensive response to these concerns, it has been proposed that we investigate topics such as the internationalization of higher education and the quality control challenges that arise throughout the process of internationalizing higher education. The dominant understanding of internationalization of higher education in China adheres to this definition, which, in a nutshell, comprises of the internalization of student sources as well as the globalization of curriculum and administration.

## The internationalization of higher education

The importance of internationalization for higher education is self-evident, even though these themes have remained insufficiently underestimated in a Chinese context. Guided by several disciplines including anthropology [24], language and communication [25], business and marketing [26], futurist studies [27], strategic leadership and pedagogy [28], internationalization challenges have emerged as a top priority for foreign universities all around the globe. Chinese colleges and universities are not bystanders to these global trends. This is in part a reaction to the changing global environment and cultural surrounds, but it is also a response to the globalisation shift itself, which, pushed to its logical conclusion, is a bottom-line development plan for any ambitious international institution. Adopting a coupling coordination model, Geng and Zhao [29] conduct a regional and temporal examination of the link between the characteristics of sustainable higher education development and the coupling coordination relationship. In addition, they present a number of concrete and actionable recommendations for ensuring the continued growth of the higher education sector. Many aspects of contemporary globalization make it necessary for institutions to modify and define the concept in accordance with their own goals; as a result, prevailing conceptions of the meta-discipline of globalization in discipline have become increasingly complex as a result of such development [30]. To a certain extent, this is especially obvious in countries where institutional internationalization is still in its early phases and where traditional Western ideas of internationalization must be studied further for their relevance in local situations. In these countries, internationalization of institutions is still in its early phases [31]. Our comprehension of the benefits and constraints associated with internationalization practice in China will be aided by the development and use of this concept in the context of such unique conditions. In addition, it is of equal significance to guarantee the high quality of such international educational practices concurrently with the process of globalization of education. Following this part, quality control in the internationalization process of higher education is then further explained in the next section, along with a review of relevant literature.

## Quality control in the internationalization process of higher education

With the commitment to enhance the world ranking of domestic higher education, transnational higher education is favoured by the Chinese government. This denotes all types of higher education study where the learners are located in a country different from the one where the awarding institution is based [32]. The term is recently interchangeably used with "cross-border education", "offshore education" and "borderless education" in related research

[33]. Transnational higher education may be conducted in different forms, such as franchise, twinning, double/joint degree, articulation, validation or virtual/distance. However, borderless education that neglects the existing borders in the delivery of transnational programs, are not included in this study.

The employment of quality control in higher education has been a global practice under the background of globalisation and internationalization [34, 35]. The integration of global markets puts modern countries under huge pressure to maintain or promote national competitiveness facing overwhelming challenges. As the knowledge base for developing potential talents, higher education has been confronting the same situation. Initiated by the state, multiple measures are taken to elevate the world ranking of domestic universities, including the quality control system, to finally enhance the efficiency and effectiveness of higher education performance [36]. Generated from the manufacturing sector, the concept of quality refers historically to consumer satisfaction, and synthesis of conformance, adaptability, innovation and continuous improvement [37]. On this basis, higher education quality is usually examined via exquisite standards, consistency with standards, adequacy of purpose, effectiveness in achieving institutional goals, and meeting stakeholders' explicit or implicit needs [38]. Since the 1980s, a series of internal reforms have reshaped the development of higher education in China, introducing privatization and marketization to accelerate the massification of education programs, which grants universities more autonomy and flexibility in university-level governance.

Quality control has become a critical issue for the orderly, healthy, and sustainable growth of China-foreign cooperative education as a result of the different disorderly circumstances and quality concerns that have developed since its emergence [39]. The majority of research have concentrated on the macro-level of government regulation and specialized quality control methods, although attention has also been made to quality control specific challenges [40, 41]. Because transnational higher education transcends national boundaries and surpasses the realm of regulation in a single country, the problem of quality control is much more difficult than quality control in a single country. This research examines the quality control of Chinese-foreign cooperative education from institutional and cultural perspectives in order to identify the underlying reasons of quality control issues.

As a consequence, in the endeavour to ensure education quality, underneath the macro efforts undertaken at the national or cross-national level, the establishment of a well-established quality control regime at the micro level of school partners demands much more academic attention and investigative effort. As the governing body in close touch with professors and students, whether a school has a solid governance structure in place directly influences the students' community. The Chinese Ministry of Education's revisions of the quality control system in higher education are primarily concerned with numbers pertaining to an educational institution's physical foundation. A well-designed campus, on the other hand, provides more than just effective and efficient educational results. Quality control can be effectively achieved only when all stakeholders' interests are considered and all stakeholders can actively participate in the governance system, and when the framework of institutional governmentality is well established and can function in promoting quality control in higher education.

Using these notions, this research seeks to use GTUC as an example by performing a case study of the quality control system from the viewpoints of institutional governmentality creation and cultural appreciation in transnational higher education. Finally, this study emphasizes the significance of high-quality international exchange and cooperation in growing China's worldwide impact and recognition of foreign education partnership with China.

## Research site and research methodology

The research methodology adopted in this study is taking a participatory action research (PAR) approach [42–44] (See Fig 1 below) and a case study method [45, 46], both of which are iconic approaches in qualitative research methodology. PAR is a rapidly growing research approach [47] in education research, thus, a proper method for educational scholars to investigate a co-developing education program with stakeholders [48]. As an important form of qualitative research, PAR through focus group interviews is an unstructured, direct, one-to-many group interview. Focus group interviews are conducted by investigators with advanced interviewing skills to reveal underlying motivations, attitudes and feelings about an issue, and are most often used in exploratory surveys, to build up subjectivity via gathering reliable pieces of evidence [49]. By gaining a detailed understanding of complex behaviour and exploring sensitive topics, the authors conducted detailed interviews with the faculty management team, frontline staff, students and many other stakeholders of the faculty at GTUC. The main function of the interviews was to obtain rich and vivid qualitative information from which to generalise and draw conclusions through the researcher's subjective and insightful analysis [50]. As for the case study, this is a major methodological approach in sociological inquiry [51, 52]. A third approach combined in this research is documentary review. This research used documentary analysis to achieve its goals. The promise of this technique is built on the measurement and comprehensive evaluation of existing records from the institution. The methodology of this research will allow for a meaningful evaluation of the school's current document on limited circumstances. The research technique is based on an assessment of current announcements, disclosures, notifications, and running profiles of this institution, as well as an investigation of several internal regulatory recordings. The many actions carried out throughout the study's development will allow the formation of landscapes with nodes showing performance, implementation of conceptual subthemes, and development linked to its internal management and quality control approaches. The above-mentioned research methodologies are scientifically valid and applicable when applied to the case of Chinese-foreign collaborative education in China.

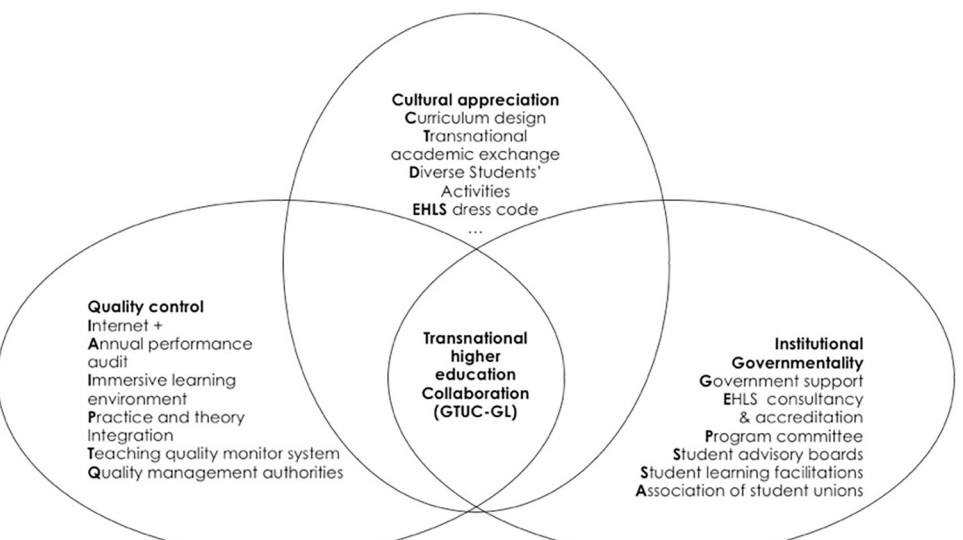

**Fig 1. A research framework for internationalization of transnational higher education collaboration in GTUC-GL, China.**

The investigation was carried out voluntarily by interview participants during an initial period of about 12 months, from January 2020 to December 2021, with the agreement of the institution and the written consent of all interviewers. For the Ethics Committee of the author's institution has accepted and fully supported the current examination of the research objects. All participants are encouraged to read the following statement in accordance with the qualitative inquiry: *I consent to participate in this qualitative understanding to transnational collaboration project in the current faculty. I concur that the response I made in the following interview to be used for academic research purposes by researchers of this scientific interpretation.* An international literature review, institutional document and policy review, and meetings with university administrators, program and course leaders, coordinators, and professional development lecturers to develop an interdisciplinary, cross-institutional case study of an internationalised curriculum are among the data resources for further analysis. The overall number of respondents in this study was 64, which included university officials, the dean of the Faculty of GL, faculty members, GL students, and a broader spectrum of Faculty of GL stakeholders. In this regard, a detailed technique or strategy for gaining an explanation of the objective and analysing the effect of modifications to that mechanism was offered by the stakeholder approach [53, 54]. This was achieved by determining who the major players or stakeholders in the system were and analysing the functions that they play in the mechanism from a strategic perspective. A definition of stakeholder that is widely recognized and accepted categorizes as stakeholder any group or individual that has the potential to influence or is affected by a development and/or the achievement of the goals of an organization. This definition applies to both the development itself and the goals themselves.

The authors mediated the focus group interviews, and the interview procedures were recorded by research assistants. Based on current research on internationalization, cultural appreciation, and quality control in collaborative international higher education cooperation, several key open-ended issues were proposed. These prepared questions maintained the flow of discussion topics and regulated the focal scope of the major subject of this research endeavour, which focused on (1) How academics working in various institutional and disciplinary settings define the idea of curricular internationalization? (2) How does the school include faculty in the process of internationalizing the curriculum in academic practices? (3) How is the quality control system developed and monitored during the implementation process? Discussions were also encouraged throughout the question and answer segment to delve into the latent meanings behind the participants' replies [55, 56]. The respondents were given the freedom to disclose their actual feelings, which was made possible by the researchers' support. The researchers attempted to provide a comfortable and free interactive environment for the respondents, in order to encourage deeper and more free communicative conversations [57].

Three rounds of focus group interviews were held in the same location to ensure dependability and credibility in qualitative educational research methods [58]. Each interview lasted around 40 minutes to an hour. The researcher audio-recorded the interviews with the permission of the participants and subsequently converted them to text using technological techniques. The three interviews' texts were combined. After acquiring the entire texts, the writers double-checked with the interviews in cases where they were dubious about the actual meaning of the interviewees. The interview inaccuracies were subsequently addressed.

Text analysis of this research project followed a further systematic investigation [59]. The original material was then transcribed as the first phase. Following each interview, the tapes were meticulously transcribed, and the source material was thoroughly examined and analysed. The second stage was to code. The current study's information was reorganized and classified, using key themes or thematic emphasizing categories. The coding results were displayed as text information and analysed using theme categories. The language of the interviews was

used to deliver the material, which helped the researchers comprehend the feelings and perspectives of the interviewees in the situation. Finally, the theme categories were compiled in text format and sent to the interviewers for a second round of validation. As a result, the authors created the graphic below to expound on the contributing elements linked to the internationalization problem in this particular situation in China.

## Research findings

Accompanied by the synthesizing of precedential literature, an overall investigation of the texts has shown the research framework for a transnational higher education collaboration in the case of GTUC-GL, as shown in previous figure. Three identifiers have been conceptualized: (1) mutual cultural appreciation; (2) institutional governmentality; and (3) quality control. Such triangle identifiers have been grounded in the focal case in this research, which is popularly cited in previous prestige pieces of literature of transnational education collaboration, for instance, culture issues [60, 61], institutional structure [62, 63], as well as the systematic process for high profile quality control [63, 64]. The research findings for each of the identifiers is then further shown and elaborated in detail in the following sections. Part of the verbatim quotes, which are greatly condensed into theoretical debates and discursive summaries, serve as testimony to corroborate the previously specified identifiers, as seen in Fig 1 above.

### Cultural appreciation in the transnational educational collaboration process of internationalization of curriculum

This section analyses the hospitality mindsets from the perspective of cultural appreciation. The internationalization of the curriculum sits at the intersection of university policy and practice and is a source of fascination and achievement for students, academic staff and university administrators. Stakeholders in the process of internationalization must be equipped with strong cultural appreciation, accepting cultural diversity, equity and inclusion [65]. Under the guidance of GTUC's internationalized education planning, with the combination of the needs of China's education and GTUC's features, GTUC-GL integrates its mission, vision and distils them down to "GLers" as its core values, which consist of *Graceful Generalist*, *Life-Long Learner*, *Responsible Rudder and Sincere Socialist*. To integrate these core values into school life, GTUC-GL starts from the following aspects to develop students into qualified high-end hospitality talents in the future.

To date, the GL teaching facility is well-formed in its lobby, guest room, banquet hall, and many other areas, and has produced a fully functioning hospitality teaching environment, complete with hotel service chain settings. The lobby is surrounded by a green planting space, a small sports field, and a sofa lounge area, providing the faculty teaching community with an open view and a pleasant ambiance in their spare time. Integrated cultures from both Switzerland and China have infiltrated the staff and students' imaginations. Such impressions were instilled in teachers and students during their initial visits to the EHLS campus, and therefore from the time they were recruited to the institution. These accomplishments did not come easily or quickly. Interviewees who were present at the initial establishment of the combined higher education partnership revealed their opinions regarding the challenges of creating an extremely international-style teaching and learning environment. The same views have been instilled in bachelor students. The academic staff and students were to embrace the hospitality traditions of both Switzerland and China. GTUC-GL's philosophy is represented in the software and hardware infrastructure, as well as the services given to its teachers and students. A genuine hotel was built to house a cooperative education program, giving a physical foundation for establishing the EHLS teaching paradigm and delivering a learning experience via

practical training in actual hotel roles, particularly from the time freshmen enter this teaching structure. This authentic hotel was built to host this co-operative education program. It gives students the ability to gather learning experiences via practical training in genuine hotel positions and serves as the physical foundation for establishing the EHL teaching paradigm. This option is particularly open to students who are just entering this educational system. This sentiment was echoed over the course of the interviews, with interviewees expressing the perspective outlined below.

> *I believe the most crucial aspect is the ladder concept of professional training programme. The curriculum is organized in a spiral progression of "practice → theory → practice → theory". We offer students with a totally realistic operational environment in which they may study the essential professional courses, allowing them to swiftly grasp the relevant information and abilities and fully realize the synchronization between professional education and industry demands.* (G1, Professor, Head of School, one of the main directors of the School, interview in December 2021).

Adopting an EHLS educational horizon [66], GTUC has built an independent hotel as an academic building, providing an infrastructure basis for students' immersive learning. GTUC-GL has a total surface area of 21,858 square meters, which is made up of 11 multimedia classrooms and 13 practical training classrooms that enable students to engage directly in the operations of a real hotel, flawlessly blending professional experience with theoretical school learning. Students begin their first semester in the faculty by cycling through 14 mockup roles in a genuine hotel environment. Further, to keep the core culture value [67] deeply rooted in every staff and student, for instance, GTUC-GL designs its school badge, makes signage of its mission, vision and core values. The faculty incorporate these cultural symbols in the faculty teaching building. GTUC-GL absorbs EHLS's talent for developing a routine of "Practice–Theory—Practice Again—Theory Again", to strengthen features of applied hospitality talents development, which are also termed as an experiential learning model [68]. The GTUC-GL, which is based on hotel facilities, draws on EHLS's successful experience in conducting practical training to divide must-have hospitality service skills into 14 positions and design related practical training courses, achieving a high fusion of teaching and operation, theory and practice. The GTUC-GL produces an official document to incorporate hospitality professional standards into university life. The GTUC-GL professors and students adhere to a formal dress code of hospitality professionals that corresponds to their curriculum, demonstrating the GLers' positive professionalism. The School also displays dress code and international business procedure signs, tacitly inspiring its pupils to be industry talents. Fig 2 depicts cultural appreciation as a dynamic interaction process between EHLS and GTUC at the institutional level. Such mutual structural appreciation suggests that active and favourable interconnectivity between different university cultures, as Wu and Pullman [66] note, enhance the credibility and are beneficial to the organization of joint international higher education collaboration.

> *The special features for this unique school are these followings: the school's financial and policy support, the calibre of the international institutions with which the school has partnered, the employment of foreign specialists, and the educational experiences of the school's management and faculty members gained during their time spent studying abroad. This function is contingent on the applied talent development model that the GTUC uses as well as the launch of the operations integration program that EHLS has implemented. This characteristic is in line with the objective of teaching practical skills, and it may be used to highlight the school's*

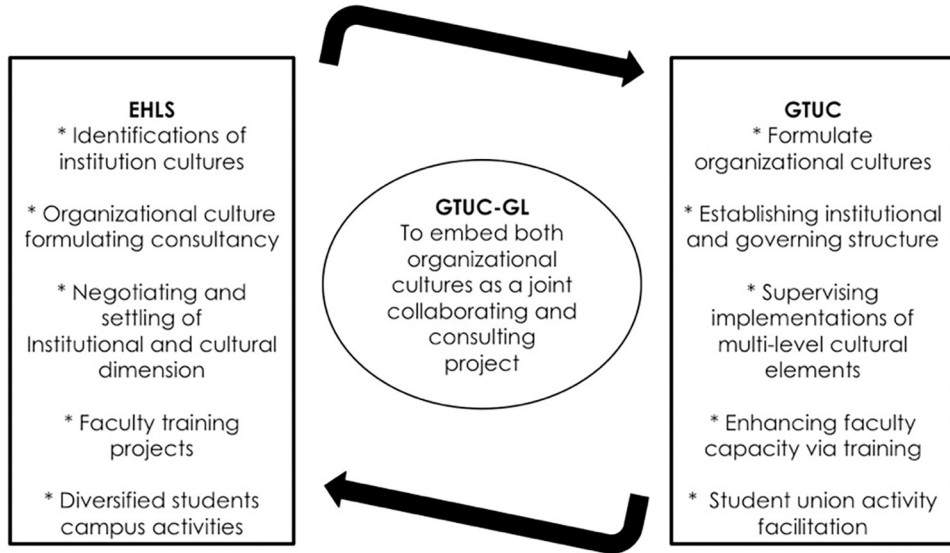

**Fig 2. Culture appreciation as a multi-facet interactive communicating mechanism.**

*training of applied talent features.* (In December 2021, an interview was conducted with G5, Professor, who was one of the supervisors at the school who was responsible for the formulation of the curriculum)

To have a more in-depth discussion of the internationalization of higher education, researchers must include the internationalization of the curriculum, which includes international-style teaching and outcome-oriented student learning [69]. Yet, the internationalization of the curriculum is little known as a concept and is evolving in practice [70]. Planning to reach the goal of internationalised learning, it must be done in the context of integrated cultures and knowledge spheres, diversified behaviour change and practices of full engagement within a transdisciplinary approach [71]. However, if academic staff do not have sufficient experience, skills or knowledge required to internationalise the curriculum, they are, thus, incapable of sufficiently engaging with fully adopting the internationalization of the curriculum. This is serving as competitive implications for universities' international strategies and student learning [72]. The intersections between disciplines, curricula, internationalization and student learning in higher education form a connected space and a glocalized curriculum that offers rich culture experiencing opportunities for students and faculty members [73]. Participants in the interview offered the following examples to illustrate how the curriculum may be made more international:

*This course of study has the intestinal fortitude to break away from the normal educational paradigm and makes an effort to integrate Eastern and Western methods of instruction. Working with EHLS, an institution that is recognized as a leader in the hotel management industry, provides a one-of-a-kind opportunity to learn from one another. For this reason, we are enthusiastic about the progress that is being made with the program. Second, the educational technique of the program is a good match with what I've learned from my own experience studying in other countries. I have high expectations for the project and look forward to acquiring the most cutting-edge knowledge from both the Western and Chinese educational systems as well as improving my general competence. I want to be able to lead the program in*

*such a way that it produces students who have a more global perspective and a more holistic approach, in addition to individuals who have professional skill and moral integrity.* (Interview conducted in June 2021 with G4, Professor, who is largely responsible for teaching courses at the school)

Cultural factors also need to be taken into examination in the quality control system. Mutual cultural appreciation in social sciences usually refers to "the dependence of a phenomenon . . . in institutional, social, cognitive or cultural terms" [74]. The faculty members of hospitality management cultivate students' comprehensive competencies through all kinds of cultural- and hospitality-related activities [75]. Besides organizing competitions of professional skills such as the "GL Cup" Sommelier Services Competition and Competition of Creative Table Napkin Folding, GTUC-GL selects outstanding students to participate in professional competitions at the national and provincial level and Young Hoteliers Summit, to transfer students from inexperienced youngsters to potential professionals with strong cooperation social responsibility mindsets [76]. Here, cultural appreciation is innovatively adopted, to refer to the integration of different cultures reflected in transnational higher education programs [77]. The school culture where all stakeholders are surrounded is shaped through the combination of features of two national cultures and that of institution partners. The vision, mission and strategy are further adopted and applied to find the origin of the cultural appreciation. By absorbing the exceptional characteristics of each side, the entity operating transnational cooperation programs can share with the whole community the ideal framework of education outcomes, which will lead to the enhancement of education quality. For faculty members, they are both hotel operators and learning facilitators, passing on the spirit of dedication, professionalism and enthusiasm to students through their proper behaviours. These student contests are some of the methodologies for students to to acquire cultural perspectives. In hotel management education, the GTUC-GL faculty provides accessible services for workers' job and life, as well as a focus on students' physical practices and mental growth.

## Research finding 2: An institutional governmentality reflected in the faculty

Institutional governmentality is a conceptual approach to public service administration study. In a similar vein, the next part employs the concept of institutional governmentality to further examine the faculty's administrative organization. Using this method, this section explores the complexities of the GTUC-GL faculty's internal institutional structure, similar to what Baxstrom [78] notes, to exemplify the structural planning and operational multiplicity. To this extent, GTUC has set up a fully functional institution to implement the original aims and scopes of the school (See Fig 3 below).

GTUC has formed a Worldwide Advisory Committee comprised of distinguished international hospitality education and industry specialists from Hong Kong Polytechnic University, the University of Surrey (UK), the University of Houston (USA), Sun Yatsen University (PRC), and the Banyan Tree Group. Each year, committee members convene to exchange ideas and provide constructive proposals in response to GTUC's difficulties and current issues, in order to assist GTUC review its growth path and goals. The GTUC-GL management team, as a GTUC school, actively integrates these yearly proposals into the innovation of its talent training and operating direction. Respondents showed a high level of satisfaction and awareness when one respondent made comments about the concept of institutional governance.

*When I was teaching in such an environment, I was able to feel the support of the institution, the support of the government, the good participation from the foreign side, the nice*

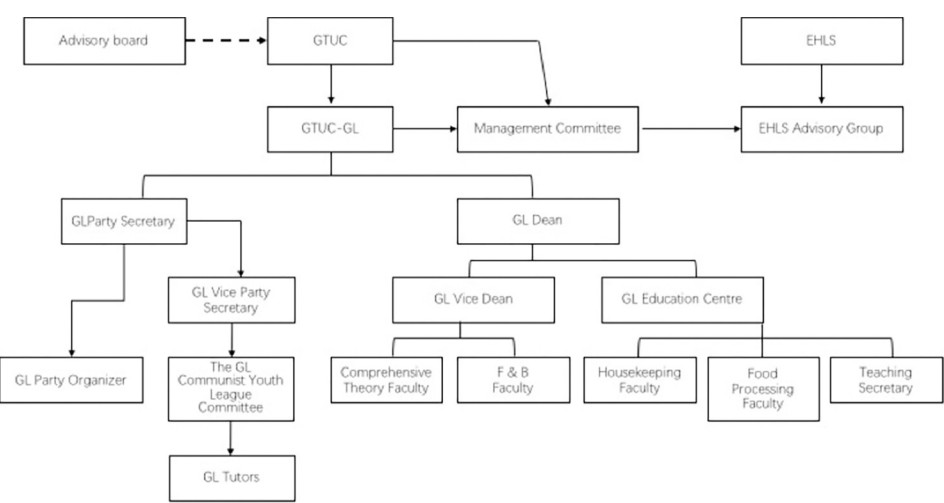

**Fig 3. Hierarchical structure for administrative management chart of GTUC-GL.**

*atmosphere of several languages being used, and so on. This endeavour has garnered a great deal of support from a number of departments throughout the institution. For instance, the department that is in charge of international cooperation has assigned a member of their staff to oversee the abroad component of the project. In terms of the English language, the institution places a strong emphasis on providing students with extensive active pre-entry English language preparation, as well as expanded language classes and other related activities. The standard living and learning environment of the students contributes positively to the students' use of the English language. These initiatives have made a substantial contribution, collectively, to the internationalization of the institution.* (E5, teacher and teaching assistant, responsible for teaching courses in second languages at the school who have experience studying abroad, interview took place in June 2021)

Furthermore, GTUC and EHLS have formed a China-foreign Cooperation Program Management Committee comprised of nine members from both sides, led by GTUC's president. The Committee meets on a regular basis to summarize and report on the school's operations, as well as to consider the school's future direction and action plan. Student responses complimented the work of the Committee and the School's teaching management team, with one student remarking,

*For student management, we have midterm and final exams, as well as feedback on instructors' teaching every weekend. Regarding instructor management, the school employs stringent instructional monitoring, biannual academic assessments, and periodic Swiss visits. This curriculum deviates from standard higher education in that it combines theory and practice in its instructional technique. In accordance with this arrangement, we complete two industrial internships over the course of two semesters in our second and fourth years, and we have cycled through over a dozen break-related responsibilities in our first year to provide culinary and lodging services for students, teachers and real visitors. Through this extensive practical industry training, we gained a comprehensive understanding of our suitability for the business, as opposed to entering our senior year knowing our professional goals.*(Interview with T5 senior in this program as of December 2021)

To guarantee teaching quality, the GTUC-GL has formed a teaching management team comprised of the vice president in charge of teaching, the director of each faculty team, and

the teaching secretary. The school has a Party branch with a secretary, a deputy secretary, and an organizer to carry out student management in accordance with GTUC regulations, assist students in resolving study and life challenges, and collect and provide feedback on students' views.

GTUC-GL analyzes teaching and learning at the conclusion of each semester to ensure that the measures are fully implemented. The school monitors instruction via reciprocal assessment between instructors and students, as well as random review by professionals. The teaching secretary compiles the assessment data and communicates them to school authorities, detailing the faults that were identified. Next a meeting of all faculty members, the school adopts and implements appropriate actions to alter the curriculum for the following academic year.

## Research finding 3: Improvement and progress in teaching quality control

This research recognized current signals of development as a quality control system, immersive learning environment, customized curriculum design, and hands-on learning chances as hotel intern managers. The Teaching Quality Monitoring and Assessment Centre at GTUC is in charge of monitoring teaching activities, analysing monitoring data, and offering comments. The key revision procedures for the curriculum and instructional materials at GTUC-GL are shown in the figure below (see Fig 4 below).

The GTUC-GL offers students an immersive learning environment in the form of a real-life mock-up hotel. An independent hotel is built to provide a genuine operating environment in order to achieve integration of education and operation. To meet the learning goals throughout the teaching process, a four-in-one model has been developed: (1) the integration of Chinese and Swiss education in one faculty; (2) the educational school teaching and a training business in one location; (3) being instructors or students as well as workers in one location; and (4) employees and consumers in one school (for faculty members and students are consuming in the teaching building). 13 distinct experimental training courses are built up to

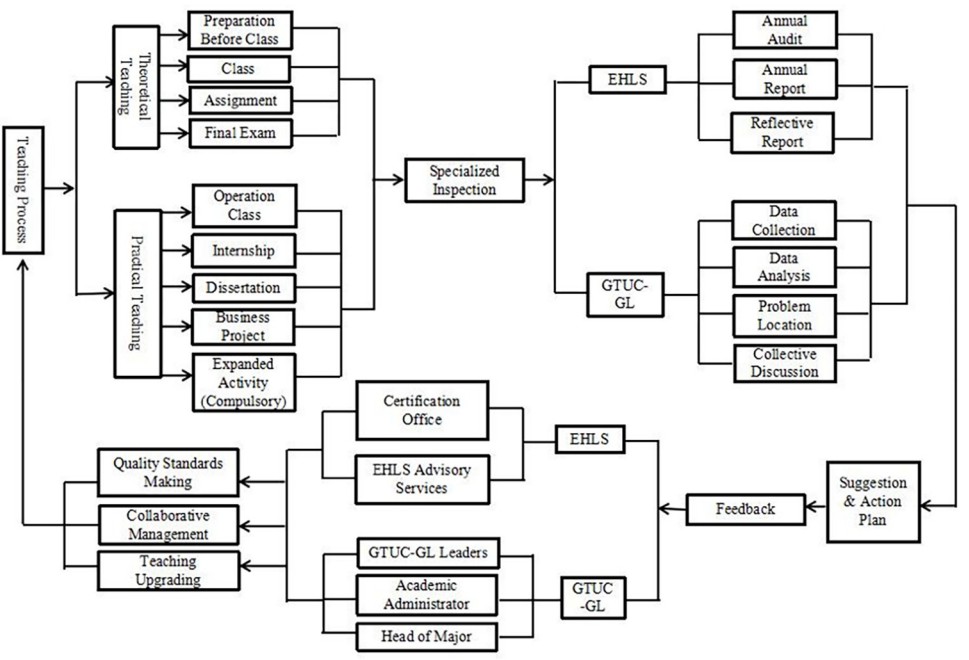

**Fig 4. Closed-loop diagram for monitoring teaching quality in GTUC-GL (summarized by authors).**

integrate professional competences with the curriculum, establishing a high degree of integration of theory and practice, as well as teaching and operation in daily routine, based on the must-have hospitality services and operation capabilities. This characteristic is clear from the respondents' comments from one of the students interviewed.

*This curriculum provides an advanced international teaching approach with a high degree of practice and theory integration, with one year of practice and three years of theory in four years, as well as two genuine industrial internships alongside the theoretical courses, according to my personal experience. At the same time, we are obliged to enter the building in operational condition. In other words, a formal professional appearance is required. We have four years of experience and a very professional mindset with this operation before we graduate. As students of the program, we think we are extremely competitive in the business and possess the diverse knowledge and professional skills necessary for success in the hospitality sector.* (T7, a sophomore in the program who was still in her first industrial internship at the time of the interview in June 2021, was interviewed)

Second, for training and hospitality professionals, a spiralling pattern for nurturing talent known as the Practice & Theory Progression model of curriculum design has been established. The curriculum is then well-structured based on a two-time spiral of Practice-Theory-Practice-Theory rotation. This innovative teaching method received a medal for the Guangxi Zhuang Autonomous Region's educational excellence. As one of the primary student activities, the GTUC-GL creates a multidimensional and comprehensive training platform comprised of Innovative Business Start-ups. Innovative business activities and start-up contests have evolved into training opportunities for students to improve their overall competencies. To some degree, these kids establish a cross-border, multidisciplinary, and cross-grade multiple collaborative innovation and entrepreneurship. Business project training techniques, shown as teacher-student co-working styles, are designed to develop students' potential for invention and entrepreneurship. Educators encourage and support students' inventiveness and entrepreneurial potential when they participate in contests involving business ventures. Similar sentiments were expressed by students during interviews:

*We host several cultural activities to encourage cultural integration, such as an annual food service etiquette competition and a wine label design competition. In addition, we provide a variety of student events, such as the school's unique Friday Night and our student-designed '520 Special Food Service' event, which entails preparing special coffee and cocktails. These activities are an excellent means of fostering cultural integration, motivation, and an atmosphere for learning.* (T6, current university student in the China-foreign collaboration program, third year, interviewed at the beginning of the school year in September 2021)

Finally, the GL institution selects students each year to participate in the "Internet+" Innovation and Entrepreneurship Competition and establishes various awards to encourage students to actively participate in such activities and improve their innovation and entrepreneurial skills, organizational and coordination skills, and so on. Students might apply for roles such as Intern Manager to work as a manager at a real hotel. Students are participants in hotel operations as well as learning actors, embracing the spirit of hospitality and enhancing their overall skill as hoteliers via immersive education. GTUC-GL has professional employees to provide all-around high-quality education and services. GTUC-GL presents EHLS's education model, incorporates GTUC's applied undergraduate program features, and offers a curriculum that blends theory and practice to prepare students to become professional hoteliers. The

School also establishes scholarships and bursaries to recognize excellent students, assist needy students in overcoming financial difficulties, and give all-around services to enhance students' campus life.

## Discussion

With the fast expansion of Chinese-foreign cooperative education, it is critical to establish and continually enhance a quality control system for Chinese-foreign cooperative education from institutional and cultural perspectives [37, 79]. Three identifiers are described in the following sections based on an examination of the interview data obtained. To get a more in-depth understanding of the data presented above, this study will continue to analyse these three identifiers in order to investigate strategies to enhance the worldwide recognition of Chinese education. Furthermore, this debate aims to represent the GTUC-GL scenario of the quality control system of this transnational higher education partnership between China and Switzerland.

### Quality control in transnational higher education collaboration benchmarked by institutional constructions

Quality control must be one of the most important parts of the internationalization process to the cooperative international higher education partnership [80, 81]. Due to the unique characteristics of international cooperation, we should not simply apply the traditional mode of quality assessment of domestic higher education, but should actively learn from the common practice of international quality control of transnational higher education, actively investigate the effective mechanism of appropriate separation of management, administration, and evaluation, and strengthen cooperation with quality control institutions and organizations of transnational higher education [82–85]. To improve and enhance the quality of China-foreign cooperative education, the government should strengthen cooperation with quality control agencies and organizations of transnational higher education exporters, as well as build an external quality assessment system with Chinese characteristics and in accordance with international standards [86]. Accreditation is often recognized as the most prevalent instrument for managing access to and quality control in the transnational higher education market. The variety of Chinese and international relationships should be reflected in accreditation. Because programs and institutions are different and unique, it is not possible to adopt a single methodology to accredit and evaluate them all. It is critical to present the various types of Chinese-foreign cooperative education, as well as the various needs and standards of Chinese-foreign cooperative education, so that the interests of Chinese-foreign cooperative education providers, students, and teachers can be better reflected through industry cooperation.

### Quality control in transnational higher education collaboration benchmarked by an effective management system

To fairly satisfy the criteria of the original purposes of its initial foundation, an efficient management system should be established for developing a co-operative education institution [87–90]. Effective quality control and risk management methods may be properly built to govern their educational practices by effective communication and trust between the cooperating two parties, as well as honouring their separate worldwide reputations. Promoting institutional internationalization would therefore be accomplished by protecting their individual credentials and co-competed course planning quality [86], as well as attaining mental agreement based on structural disparities. In terms of quality control management models, the shift from

management to governance should be accomplished gradually. It is critical to specify the responsibilities of the government, school operators, and social intermediate organizations throughout the quality control process. Each stakeholder in a China-foreign cooperation project should clarify the respective function by forming a synergy of corporate social responsibility and coordination by constructing a scientific, rational, and effective quality control system [91, 92]. The national governmental oversight authority should concentrate on the aims, motives, and goals of Chinese-foreign cooperative education. The government's quality control obligations at the local level are mostly tied to the execution of national policies and regulations. The institution is the primary topic of quality control since it is the primary supplier of education, and its quality risk control is based on the degree of satisfaction of students with the educational materials they get and the educational delivery method. From a cultural standpoint, the necessary core components for a good transnational higher education partnership are the acknowledgment of heterogeneous knowledge and reciprocal acceptance of cultural appreciation. In accordance with this cultural viewpoint, the pursuit of complementary capabilities and interests should be emphasized in the provision of cultural integration to both local and foreign cultures, as expressed in policy decisions and the quality control process. In terms of academic quality, it is critical to stress the comparison of transnational education programs with national programs. In the course of the growth of China-foreign cooperative education, several quality criteria have been devised., such as the *Guidelines for the Evaluation of China-foreign Cooperative Education (for trial implementation)* and the *Guidelines for the Selection of Exemplary China-foreign Cooperative Education Projects (for trial implementation)* [93]. Despite the fact that these standards are still in the prototype level, they have a significant impact on the cultural attitude of the faculties and students as a whole. To further enrich and strengthen quality control standards, a successful international exchange and cooperation project would be able to expand and secure the healthy growth of China-foreign cooperative education.

## Conclusion

The present study analyses concern of internationalization, cultural appreciation, and institutional governmentality for quality control in the instance of GTUC-GL as a transnational higher education partnership by employing participatory action research and a case study methodology. GTUC-GL presents its own applicable GL model of developing hospitality skills in collaboration with EHLS, in order to build a highlighted talent-cultivating culture. The GTUC-GL is devoted to be a leading institution in discovering and disseminating applied talent training methodologies, a cultivator of future high-end industrial talents, and a catalyst for school-enterprise collaboration. The program's goal is to produce high-quality graduates while also offering a plethora of experience opportunities for the development of a distinguished undergraduate program.

The findings of this work give valuable insights for tourism and hospitality researchers and faculty members in the institutional structure, as well as career counselling for scholars interested in Chinese international partnership for higher education. The study adds to current academics not only via its conceptual treatment of cultural concerns, governmentality, and quality control in this arena, but also through its application in practice. According to the findings, the quality control system should identify the relative functions and responsibilities of all stakeholders, transition from management to institutional governmentality, and progressively strengthen quality control methods. Countermeasures such as enhancing and increasing quality standards in its mutual cultural appreciation from both sides must also be incorporated.

Because of the current constraints, the results and comments should be interpreted with future discretions. To begin, this research solely examines into one example through two

collaborated institutions. Future study should investigate these research themes in broader, more substantial cases, using more diverse research approaches, such as a quantitative approach, or with a broader spectrum of stakeholders. Second, although this study adds to the literature by establishing a research framework and approach to the issue via three conceptual notions, it is advised that future studies compare these conceptualizations with other forms of philosophical thought Future research may help to enhance our knowledge of how to build a high-quality transnational higher education cooperation throughout the globe, as well as discover a more fundamental philosophic conceptualization on this education collaboration issue.

## Author Contributions

**Conceptualization:** Jinsheng (Jason) Zhu, Shushu Wang.

**Data curation:** Jinsheng (Jason) Zhu, Shushu Wang.

**Formal analysis:** Jinsheng (Jason) Zhu.

**Funding acquisition:** Jinsheng (Jason) Zhu.

**Investigation:** Jinsheng (Jason) Zhu, Shushu Wang.

**Methodology:** Jinsheng (Jason) Zhu.

**Project administration:** Jinsheng (Jason) Zhu.

**Resources:** Jinsheng (Jason) Zhu.

**Software:** Jinsheng (Jason) Zhu.

**Supervision:** Jinsheng (Jason) Zhu.

**Validation:** Jinsheng (Jason) Zhu, Shushu Wang.

**Visualization:** Jinsheng (Jason) Zhu.

**Writing – original draft:** Jinsheng (Jason) Zhu.

**Writing – review & editing:** Jinsheng (Jason) Zhu, Shushu Wang.

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
