## [Decision Letter · Decision Letter 0]

31 May 2022

PONE-D-22-11647Internationalization, cultural embeddedness and institutional governmentality for quality control in transnational higher education cooperation: An empirical assessmentPLOS ONE

Dear Dr. Zhu,

Thank you for submitting your manuscript to PLOS ONE. After careful consideration, we feel that it has merit but does not fully meet PLOS ONE’s publication criteria as it currently stands. Therefore, we invite you to submit a revised version of the manuscript that addresses the points raised during the review process.

Your choice of topic and research demonstrate a firm commitment to contribute to the better understanding of TNE in China. It is an important project, which may offer a fresh insight into the major items on your agenda, i.e. internationalization, cultural embeddedness and the question of quality control at institutional level. Although Reviewer 2 was suggesting a full green light, I agree more with Reviewer 1 and the detailed criticism provided over there.

Please, revise your manuscript thoroughly in particular with regard to your theoretical framework, research questions and empirical work. As for the literatiure review, consult more recent pieces published in the relevant fields, also possibly with a wider look at internationalization and the geopolitics of higher education. Reviewer 2 raises several serious problems about your findings and how you interpret your data.

We look forward to receiving your revised manuscript.

Kind regards,

István Tarrósy, PhD

Academic Editor

PLOS ONE

Journal Requirements:

3. Please ensure that you include a title page within your main document. We do appreciate that you have a title page document uploaded as a separate file, however, as per our author guidelines (http://journals.plos.org/plosone/s/submission-guidelines#loc-title-page) we do require this to be part of the manuscript file itself and not uploaded separately.

Reviewers' comments:

Reviewer's Responses to Questions

**Comments to the Author**

1. Is the manuscript technically sound, and do the data support the conclusions?

Reviewer #1: Partly

Reviewer #2: Yes

2. Has the statistical analysis been performed appropriately and rigorously? 

Reviewer #1: N/A

Reviewer #2: Yes

3. Have the authors made all data underlying the findings in their manuscript fully available?

Reviewer #1: No

Reviewer #2: Yes

4. Is the manuscript presented in an intelligible fashion and written in standard English?

Reviewer #1: No

Reviewer #2: Yes

5. Review Comments to the Author

Reviewer #1: There is no doubt that there is growing interest among higher education institutions across countries of the world to develop the transnational higher education setting to reflect evolving global needs. The current study – “Internationalization, cultural embeddedness and institutional governmentality for quality control in transnational higher education cooperation: An empirical assessment” is very important to the development of TNE in China. While I commend the authors for the time and effort invested in the study, I think that there are additional work that need to be done in all the sections in order to bring the research to the journal’s standards. First, there is a lack of a clear theoretical underpinning of the study. I do not think that the authors’ discussions on internationalization and transnational higher education provide adequate information on the theoretical foundation. Secondly, although the introduction sets a very good pace for the delivery of very informative discussion on TNE and what the authors term as cultural embeddedness and institutional governmentality for quality control, the literature review and empirical sections were short of fulfilling those expectations. The major problem identified with the research findings is the lack of “thick data” or “participants’ narrative” to inform the findings. Summarily, although the current study is very important to the field of transnational higher education, I do not think that the authors have demonstrated enough evidence to show what new knowledge emerges from their study. I have provided some additional comments below for their attention.

Literature review

I think that the literature review section needs to be improved substantially to provide readers with better understanding of internationalization and transnational higher education. In its current form, the literature review section reads like an extension of the background section. The authors should provide answers to some vet important questions. For example: What are the theoretical foundations of internationalization and transnational higher education? What is already known in the field? What are the existing gaps in the concepts? Etc. unfortunately, these very important information cannot be found in the current text.

Research site and research methodology

The authors have not provided adequate information concerning the population of their study, the sampling technique adopted to select participants for the study and how the questions were developed. I read the literature review section again to find out how the authors developed their research questions but I could not gather adequate information on that. Also information on the process used to analyze the data is very scanty. Seeing that that authors followed a systematic investigation process, readers should be informed about how the codes were developed into the three main themes. This further connects to my earlier comment concerning the gap between the questions and the theories/concepts used by the authors.

Research findings

I struggled to identify how the narratives of the focus group participants were captured in the findings section. Did the researchers decide to abbreviate the contributions of participants? I suggest that the authors take a second look at the content and structure of the research findings section and provide readers with the rich text that reveal the voices of study participants. The discussion section also relies heavily on the secondary data analysed rather than the findings from the narratives of participants.

Reviewer #2: The topic of this article is very relevant from the viewpoint of internationalization trends in higher education. The research methodology (PAR, case study) adopted in this study is proper. Conclusions are clear and well established. There are some typing mistakes (e.g. lowercase instead of capital letters) left in the text. Please correct them.

6. PLOS authors have the option to publish the peer review history of their article (what does this mean?). If published, this will include your full peer review and any attached files.

Reviewer #1: **Yes: **Dr. Yaw Owusu-Agyeman

Reviewer #2: No

---

## [Author Response · Author response to Decision Letter 0]

22 Jun 2022

Cover letter in R1 submission

June 2022

Dear Editor-in-Chief:

This article has been updated in response to the feedback provided by the editors and reviewers. Your encouraging email to request a major revision of the article with the code PONE-D-22-11647 "Internationalization, cultural appreciation, and institutional governmentality for quality control in transnational higher education cooperation: An empirical assessment" is very much appreciated. We are really appreciative of your editorial board's decision to provide us with the chance to resubmit our work. We have given the remarks made by the reviewers and your own ideas careful consideration. We have every reason to believe that this piece of writing will be effective in appealing to a wide audience of possible readers. 

We have also added, in accordance with your specific demand, the file of "point-to-point response to the reviewers" in the article submission system. This file includes a comprehensive explanation of all of the modifications that were made in this R1 version. We have high hopes that following this significant revision, you and your prestigious blind reviews will be able to once again provide us with favorable feedbacks and remarks that are encouraging. 

We also add one significant originator who began the research concept of the present work, Miss Shushu WANG, in this submission. She is included as the corresponding author in this resubmission procedure. In the interim, we have accompanied this submission with a Change to Authorship form that is formatted in accordance with PloS One. We really hope that this adjustment does not cause you undue stress. 

We hope that everything works out well for you in life! We are grateful to you for taking my concerns into account.

Thank you for your kind consideration!

With sincere regards, 

Jinsheng (Jason) ZHU, PhD / Shushu WANG 

Guilin Tourism University

Regarding the response you provided in the email correspondence, we will address your remarks in the next section of the current part of the cover letter below with bold blue letters shown below.

Dear Dr. Wang,

We've checked your submission and before we can proceed, we need you to address the following issues:

1. Thank you for updating your data availability statement. You note that your data are available within the Supporting Information files, but no such files have been included with your submission. At this time, we ask that you please upload your minimal data set as a Supporting Information file, or to a public repository such as Figshare or Dryad. 

Please also ensure that when you upload your file you include separate captions for your supplementary files at the end of your manuscript.

As soon as you confirm the location of the data underlying your findings, we will be able to proceed with the review of your submission.

Response to the editor: Thank you for responding so quickly to the data availability statement. We believe it is incorrect to add "supporting information file" in the statement section. Because the submission has no accompanying information file. Therefore, we removed "Supporting Information File" from the statement for this resubmission. We regret for causing you this confusion in this manner.

2. Please amend your authorship list in your manuscript file to include all the authors.

Response to the editor: We've provided complete author information in the manuscripts (clear version and the file with track changes).

3. Please amend your list of authors on the manuscript to ensure that each all authors re linked to an affiliation.

Response to the editor: In the updated manuscripts, we included all of the relevant affiliation information.

4. Please provide additional details regarding participant consent. In the Methods section, please ensure that you have specified (1) whether consent was informed and (2) what type you obtained (for instance, written or verbal). If your study included minors, state whether you obtained consent from parents or guardians. If the need for consent was waived by the ethics committee, please include this information.

Response to the editor: On page 8, you'll find further information that we've supplied about participant consents. For example, on page 8, under the heading "Methods," we put the statements that are shown below. “I consent to participate in this qualitative understanding to transnational collaboration project in the current faculty. I concur that the response I made in the following interview to be used for academic research purposes by researchers of this scientific interpretation.”

Response to the editor: Thank you for your comment. We have made revisions to the funding information accordingly. Some of the funding do not have a grant number. 

6. Thank you for stating the following financial disclosure: 

Response to the editor: 

The following phrases are included in the article's acknowledgment section. “In the article acknowledgement, we state the following words. “This article is part of academic achievements of first-class universities and disciplines in tourism management discipline (project) in Guangxi, China. The corresponding author has also been participating in research projects supported by Guilin Tourism University-China ASEAN Research Centre. This research project is financially supported by Guangxi Tourism Vocational Education Teaching Steering Committee - 2021 Tourism Vocational Education Research Project on Teaching Reform in Tourism Education (2021LYHZWZ001).”

We will be explaining in detail here. The “first-class universities and disciplines in tourism management discipline (project) in Guangxi, China” was a project of the university that the authors are working in. The Guilin Tourism University-China ASEAN Research Centre is one of the research centers that the authors Dr. Jinsheng (Jason) Zhu are working with. The “Guangxi Tourism Vocational Education Teaching Steering Committee - 2021 Tourism Vocational Education Research Project on Teaching Reform in Tourism Education” is one of the research projects undertaken by Dr. Jinsheng (Jason) Zhu. Here we would like to restate that both the authors are working and taking salaries in Guilin Tourism University, the institution we disclosed in the author title page. However, “The university and the funders had no role in study design, data collection and analysis, decision to publish, or preparation of the manuscript”.

We've returned your manuscript to your account. Please resolve these issues and resubmit your manuscript within 21 days. If you need more time, please email the journal office at plosone@plos.org. We are happy to grant extensions of up to one month past this due date. If we do not hear from you within 21 days, we will withdraw your manuscript.

Please log on to PLOS Editorial Manager at https://www.editorialmanager.com/pone/ to access your manuscript. You will find your manuscript in the 'Submissions Sent Back to Author' link under the New Submissions menu. Be sure to remove your previous manuscript file if you are uploading a new file in response to these requests. After you've made the changes requested above, please be sure to view and approve the revised PDF after rebuilding the PDF to complete the resubmission process.

We are requesting these changes to comply with the PLOS ONE submission guidelines (https://journals.plos.org/plosone/s/submission-guidelines). Please note that we won't send your manuscript for review until you have resolved the above requests. 

Thank you for submitting your work to PLOS ONE and supporting our mission of Open Science.

Kind regards,

Richard Ibañez Dilla

PLOS ONE

Once again, thank you for your kind consideration!

With sincere regards, 

Jinsheng (Jason) ZHU, PhD / Shushu WANG 

Guilin Tourism University

---

## [Decision Letter · Decision Letter 1]

25 Jul 2022

PONE-D-22-11647R1Internationalization, cultural appreciation and institutional governmentality for quality control in transnational higher education cooperation: An empirical assessmentPLOS ONE

Dear Dr. Wang,

Thank you for submitting your manuscript to PLOS ONE. After careful consideration, we feel that it has merit but does not fully meet PLOS ONE’s publication criteria as it currently stands. Therefore, we invite you to submit a revised version of the manuscript that addresses the points raised during the review process.

I was happy to see that the original submission was improved, but still more thorough work needs to be done, especially, as Reviwer 1 also underscores, the data analysis and the results section must be improved. Please, also take sufficient time to compose your sound rebuttal to all the questions, critical remarks articulated by the reviewers. If these are then accepted, I can support the acceptance of your article for publication.==============================

We look forward to receiving your revised manuscript.

Kind regards,

István Tarrósy, PhD

Academic Editor

PLOS ONE

Journal Requirements:

Reviewers' comments:

Reviewer's Responses to Questions

**Comments to the Author**

1. If the authors have adequately addressed your comments raised in a previous round of review and you feel that this manuscript is now acceptable for publication, you may indicate that here to bypass the “Comments to the Author” section, enter your conflict of interest statement in the “Confidential to Editor” section, and submit your "Accept" recommendation.

Reviewer #1: (No Response)

Reviewer #2: All comments have been addressed

2. Is the manuscript technically sound, and do the data support the conclusions?

Reviewer #1: Partly

Reviewer #2: Yes

3. Has the statistical analysis been performed appropriately and rigorously? 

Reviewer #1: Yes

Reviewer #2: Yes

4. Have the authors made all data underlying the findings in their manuscript fully available?

Reviewer #1: No

Reviewer #2: Yes

5. Is the manuscript presented in an intelligible fashion and written in standard English?

Reviewer #1: No

Reviewer #2: Yes

6. Review Comments to the Author

Reviewer #1: I must commend the authors for the hard work in revising the current manuscript. While I have seen that the authors have provided some additional information to the original version of their work, I think that there are some outstanding issues that needs to be resolved. First the authors should avoid abbreviations. For example on page 4 (literature review, line 1) the authors indicate that, “before delving into the challenges posed by the globalization of higher education, we'd want to….” The sentence should read as, “before delving into the challenges posed by the globalization of higher education, we would want to…..” Secondly, information on the process used to analyze the data is very scanty. Seeing that that authors followed a systematic investigation process, readers should be informed about how the codes were developed into the three main themes. This comment was contained in my earlier review report but they have not been addressed in the revised study. Lastly I still do not see an improvement in the results section where the narratives of the focus group participants are captured in the findings section.

Reviewer #2: The authors have successfully managed to revise their paper according to the recommendations. The revised sections (Literature Review, The internationalization of higher education, Research site and research methodology) have made this research article more suitable for publication from a theoretical and empirical point of view. It is also acceptable that the authors have only provided a summary of thoughts contributed by the interviewed participants. This does not reduce the academic quality of this paper. It is understandable to take into account the word limit. This paper will enrich the academic literature dealing with the internationalization of higher education with a special focus on quality control. I highly recommend the authors to continue their research and extend it to other Chinese universities which have a transnational cooperation in the field of tourism.

7. PLOS authors have the option to publish the peer review history of their article (what does this mean?). If published, this will include your full peer review and any attached files.

Reviewer #1: No

Reviewer #2: No

---

## [Author Response · Author response to Decision Letter 1]

9 Aug 2022

I was happy to see that the original submission was improved, but still more thorough work needs to be done, especially, as Reviwer 1 also underscores, the data analysis and the results section must be improved. Please, also take sufficient time to compose your sound rebuttal to all the questions, critical remarks articulated by the reviewers. If these are then accepted, I can support the acceptance of your article for publication.

Thank you so much, Dr. István Tarrósy, for your encouragement and consideration for the publication of the current paper. Your encouragement helps us a lot in the process of drafting, revising and final publication of this academic work, which we believe will be contributing to the topic of transnational higher education collaborations, quality guarantee, as well as cultural connectedness. We also believe that this paper will get high citations in the near future!

In the following response to the reviewers, we started to response to the reviewer from page 5, starting from the comments of the first reviewer.

Looking forward to hearing from you!

Yours, Wang

Review Comments to the Author

Reviewer #1: I must commend the authors for the hard work in revising the current manuscript. While I have seen that the authors have provided some additional information to the original version of their work, I think that there are some outstanding issues that needs to be resolved. First the authors should avoid abbreviations. For example on page 4 (literature review, line 1) the authors indicate that, “before delving into the challenges posed by the globalization of higher education, we'd want to….” The sentence should read as, “before delving into the challenges posed by the globalization of higher education, we would want to…..” Secondly, information on the process used to analyze the data is very scanty. Seeing that that authors followed a systematic investigation process, readers should be informed about how the codes were developed into the three main themes. This comment was contained in my earlier review report but they have not been addressed in the revised study. Lastly I still do not see an improvement in the results section where the narratives of the focus group participants are captured in the findings section.

Response to reviewer #1：Thank you so much for your comments. We are strictly considering your comments and revised the manuscript according to your criticisms point-by-point here below.

1. We have revised the unappropriated abbreviations according to your suggestion. 

2. With regard to the investigative material, we have thoughtfully selected some of the obtained original interview data for inclusion in the manuscript, which resulted in an increase of more than one thousand words to the total word count. This does not constitute a violation of the standards set out by the prestigious publication Plos One since they do not take into account the word count.

3. In the beginning, as a result of our interview that took a full year to complete from the voice recording to the transcription to the words, we gathered a total of around fifty thousand words. As a result of this, we came to the conclusion that your responses are of the highest significance. As a consequence of this, we decided to use interview data to complement the key arguments presented in each area of the article, most notably the results section. Thus, the result section turned out to be the current form. We hope you appreciate its intellectual worth in this run of review. 

We are really grateful for your views and intellectual insights. 

Reviewer #2: The authors have successfully managed to revise their paper according to the recommendations. The revised sections (Literature Review, The internationalization of higher education, Research site and research methodology) have made this research article more suitable for publication from a theoretical and empirical point of view. It is also acceptable that the authors have only provided a summary of thoughts contributed by the interviewed participants. This does not reduce the academic quality of this paper. It is understandable to take into account the word limit. This paper will enrich the academic literature dealing with the internationalization of higher education with a special focus on quality control. I highly recommend the authors to continue their research and extend it to other Chinese universities which have a transnational cooperation in the field of tourism.

Response to reviewer #2：Thank you so much for your complements! We will be very very willing to extend our research to a wilder range of scope in the near future. We hope to work with you in the near future, haha!

---

## [Decision Letter · Decision Letter 2]

8 Sep 2022

Internationalization, cultural appreciation and institutional governmentality for quality control in transnational higher education cooperation: An empirical assessment

PONE-D-22-11647R2

Dear Dr. Wang,

We’re pleased to inform you that your manuscript has been judged scientifically suitable for publication and will be formally accepted for publication once it meets all outstanding technical requirements.

Kind regards,

István Tarrósy, PhD

Academic Editor

PLOS ONE

Additional Editor Comments (optional):

As long as all the comments of the reviewers have been addressed neatly, which they (especially Reviewer 1) also accepted, and I can also see that the manuscript has improved a lot, I wish to support the publication of the paper in the journal.

Reviewers' comments:

Reviewer's Responses to Questions

**Comments to the Author**

1. If the authors have adequately addressed your comments raised in a previous round of review and you feel that this manuscript is now acceptable for publication, you may indicate that here to bypass the “Comments to the Author” section, enter your conflict of interest statement in the “Confidential to Editor” section, and submit your "Accept" recommendation.

Reviewer #1: All comments have been addressed

2. Is the manuscript technically sound, and do the data support the conclusions?

Reviewer #1: Yes

3. Has the statistical analysis been performed appropriately and rigorously? 

Reviewer #1: Yes

4. Have the authors made all data underlying the findings in their manuscript fully available?

Reviewer #1: Yes

5. Is the manuscript presented in an intelligible fashion and written in standard English?

Reviewer #1: Yes

6. Review Comments to the Author

Reviewer #1: I must congratulate the authors for the hard work. The revised manuscript looks good. I wish them all the best with their future research.

7. PLOS authors have the option to publish the peer review history of their article (what does this mean?). If published, this will include your full peer review and any attached files.

Reviewer #1: **Yes: **Yaw Owusu-Agyeman

---

## [Editor Report · Acceptance letter]

12 Sep 2022

PONE-D-22-11647R2 

Internationalization, cultural appreciation and institutional governmentality for quality control in transnational higher education cooperation: An empirical assessment 

Dear Dr. Wang:

I'm pleased to inform you that your manuscript has been deemed suitable for publication in PLOS ONE. Congratulations! Your manuscript is now with our production department. 

Kind regards, 

on behalf of

Dr. István Tarrósy 

Academic Editor

PLOS ONE